# Pneumococcal Vaccination in Adults: A Narrative Review of Considerations for Individualized Decision-Making

**DOI:** 10.3390/vaccines11050908

**Published:** 2023-04-27

**Authors:** Kay Choong See

**Affiliations:** Department of Medicine, National University Hospital, Singapore 119228, Singapore; kaychoongsee@nus.edu.sg

**Keywords:** immunity, herd, immunity, mucosal, immunogenicity, vaccine, pneumococcal vaccines, serogroup

## Abstract

Pneumococcal disease remains one of the major causes of severe disease in both children and adults. Severe disease may be prevented by pneumococcal polysaccharide and conjugate vaccines, which currently cover more than 20 serotypes. However, unlike routine pneumococcal vaccination in children, guidelines promote only limited pneumococcal vaccination in adults, and do not cater for decision-making for individual patients. In this narrative review, considerations for individualized decision-making are identified and discussed. This review identifies and discusses considerations for individualized decision-making, including the risk of severe disease, immunogenicity, clinical efficacy, mucosal immunity, herd immunity, concomitant administration with other vaccines, waning immunity, and replacement strains.

## 1. Introduction

Pneumococcal disease is caused by a gram-positive pathogenic diplococcus bacterium called *Streptococcus pneumoniae*. The bacterium possesses a surface polysaccharide capsule that inhibits ingestion and killing by leukocytes. Each capsular polysaccharide is antigenically unique and represents a different serotype. More than 90 serotypes of *S. pneumoniae* exist [1], with varying degrees of virulence. For instance, among Danish pneumococcal meningitis patients, serotype 3 was associated with a higher mortality rate than serotype 1 (23% vs. 3%) [2]. In another study, certain serotypes (e.g., 1, 5, 7) were up to 60 times more likely to be found in patients with invasive disease, compared with other serotypes (e.g., 6A, 15) that were typically found in asymptomatic individuals [3].

Human-to-human transmission of *S. pneumoniae* occurs via intimate contact or aerosol exposure. Infection by *S. pneumoniae* can lead to asymptomatic nasopharyngeal carriage, localized respiratory disease (sinusitis, pneumonia, otitis media) or severe systemic disease. Pneumococcal disease remains one of the major causes of severe disease in both children and adults, especially when invasive pneumococcal disease (i.e., isolation of *S. pneumoniae* from a normally sterile site, and associated with bacteremic pneumonia, meningitis, endocarditis, or arthritis) or severe non-bacteremic pneumonia occur. Annually, in the United States, invasive pneumococcal disease has an incidence of 10.6/100,000, while pneumococcal pneumonia accounts for 900,000 cases and 400,000 hospitalizations [4]. Such severe pneumococcal infections cause substantial morbidity and mortality, with mortality from pneumococcal pneumonia ranging from 5 to 7% [4,5]. Additionally, *S. pneumoniae* is the leading bacterial cause of pneumonia globally [6], and is responsible for about a third of bacterial community-acquired pneumonia in the United States [7].

Severe pneumococcal disease may be prevented by pneumococcal vaccines. In severe infection, proinflammatory cytokines produced by systemic infections may trigger endothelial dysfunction, atheroma instability, plaque rupture, and myocardial infarction [8]. By preventing severe disease, pneumococcal vaccination thus also has cardioprotective effects, reducing the risk of myocardial infarction and mortality [9].

Like for children, vaccination in adults aims to prevent severe disease. However, unlike routine pneumococcal vaccination in children, guidelines only promote routine pneumococcal vaccination in older adults aged ≥ 65 years. For other adults, pneumococcal vaccination is recommended only for those who are at higher risk of severe disease due to chronic medical conditions, anatomical risk for meningitis, or immunocompromise [10]. These recommendations leave situations where the indication for vaccination and the choice of vaccine remain unclear for individuals (e.g., young adult patients with history of invasive pneumococcal disease but without any known risk factors). In addition, vaccine recommendations adopt a societal perspective and may not recommend vaccination due to cost. For instance, although the risk for pneumococcal disease starts to increase from 50 years of age, the risk is only sufficiently high from 65 years of age for vaccination to be cost-effective in generally healthy adults [10].

In this review, considerations for individualized decision-making are therefore identified and discussed. Knowing these considerations will allow individualized decision-making as societal perspectives adopted by guidelines do not always apply to individual patients. Apart from risk of severe disease, these considerations include immunogenicity, clinical efficacy, mucosal immunity, herd immunity, concomitant administration with other vaccines, waning immunity, and replacement strains.

## 2. Methods

A search of PubMed^®^ (pubmed.ncbi.nlm.nih.gov, accessed on 25 March 2023) of contemporary trials over the past 5 years from 2018–2023 was performed using the term “pneumococcal vaccination randomized trials”. This was done to update the author’s personal library of articles. Articles relevant to the considerations covered in this narrative review were included.

## 3. Risk of Severe Pneumococcal Disease in Adults

Not surprisingly, patients at risk of invasive pneumococcal disease or severe nonbacteremic pneumonia include those at the extremes of age (age < 2 or ≥ 65 years), and those with chronic medical conditions (e.g., chronic cardiovascular, pulmonary, renal, liver disease [11]; diabetes mellitus [12,13]; alcohol abuse; smoking [14]; malignancy [15]; immunosuppressive conditions). Immunocompromised states increase risk of the pneumococcal disease, and these states can be identified using biomarkers such as neutropenia, low CD4 cell count, and hypogammaglobulinemia. Functional or anatomic asplenia due to sickle cell disease or splenectomy predisposes to recurrent pneumococcal disease. These risk groups also suffer from increased mortality from pneumococcal bacteremia [16,17].

As such, these patients may specially benefit from prevention of infection. For instance, in patients with cardiovascular disease (established heart failure, coronary disease, and cerebrovascular disease) or at a very high cardiovascular risk, a systematic review and pooled analysis of 5 observational studies showed that pneumococcal vaccination was associated with a 22% decrease of all-cause mortality (hazard ratio 0.78, 95% confidence interval 0.73–0.83) [18].

## 4. Pneumococcal Vaccines and Serotype Coverage

Humoral immunity is key to the host defence against *S. pneumoniae.* Vaccination should induce antibodies directed to the capsular polysaccharide, and pneumococcal vaccines use capsular polysaccharides from pneumococcal serotypes that commonly cause invasive disease. The response to pneumococcal vaccine in adults is measured by the rise in antibody levels (immunoglobin G) or serum opsonophagocytic activity titres (the maximum dilution at which the test serum shows an opsonizing/phagocytic effect using peripheral blood leukocytes as effector cells) after vaccine administration.

Two types of pneumococcal vaccine are available for clinical use: the pneumococcal polysaccharide vaccine (PPSV) and the pneumococcal conjugate vaccine (PCV). PPSV consists of partially purified pneumococcal capsular polysaccharides, with the only available formulation being PPSV23. PPSV23 contains 23 pneumococcal polysaccharides derived from the 23 serotypes that most frequently caused invasive pneumococcal disease in United States adults in the 1980s [19]. In a systematic review and pooled analysis of 21 studies involving 826,109 adult participants, PPSV23 showed significant protection against invasive pneumococcal disease [20].

In contrast to PPSV23, PCVs are inactivated vaccines that consist of type-specific pneumococcal capsular polysaccharides conjugated to a carrier protein or proteins. This carrier protein elicits a T cell-dependent memory response, which improves vaccine effectiveness during the first two years of life. Immunogenicity does not seem greatly different between the two vaccine types [21,22], though some studies show that PCV13 elicits better immune response among adults compared to PPSV23 [23]. Nonetheless, unlike PPSV23, PCV vaccines induce mucosal immunity, which in turn reduces nasal carriage of *S. pneumoniae* and human-to-human transmission of vaccine-serotypes. A variety of PCVs exist with numbers indicating the number of pneumococcal capsule types included in the vaccine, resulting in varying extents of serotype coverage:PCV13 serotype coverage: 1, 3, 4, 5, 6A, 6B, 7F, 9V, 14, 18C, 19A, 19F, 23FPCV15 serotype coverage: 22F and 33F, in addition to PCV13 serotypesPCV20 serotype coverage: 8, 10A, 11A, 12F, 15B, 22F, and 33F, in addition to PCV13 serotypes [24,25,26]PCV21 serotype coverage (not yet commercially available): 3, 6A, 7F, 8, 9N, 10A, 11A, 12F, 15A, 15C, 16F, 17F, 19A, 20, 22F, 23A, 23B, 24F, 31, 33F, 35B. These serotypes contribute to 74–94% of invasive pneumococcal disease in adults aged ≥65 years [27]. Eight of these pathogenic pneumococcal polysaccharides (15A, 15C, 16F, 23A, 23B, 24F, 31, and 35B) are not included in any currently licensed pneumococcal vaccines. Among 508 adults in a Phase 2 study (71% 50–64 years and 29% ≥65 years of age), PCV21 was non-inferior to PPSV23 for 12 shared serotypes and superior to PPSV23 for 9 unique serotypes in PCV21 [27]PCV23 serotype coverage (not yet commercially available): 2, 8, 9N, 10A, 11A, 12F, 15B, 17F, 20, 22F, and 33F, in addition to PCV13 serotypes (except for 6A). Serotypes 9N, 17F, and 20 are included in PPSV23 and are absent from PCV20PCV24 serotype coverage (not yet commercially available): 1, 3, 4, 5, 6A, 6B, 7F, 9V, 14, 18C, 19A, 19F, 23F (shared PCV13 serotypes) plus 2, 8, 9N, 10A, 11A, 12F, 15B, 17F, 22F, and 33F (shared PPSV23 serotypes) and 20B. This vaccine also incorporates a fusion protein consisting of two highly conserved pneumococcal proteins (sp1500 and sp0785) that may help protect against both vaccine and nonvaccine serotypes [28]

When both PPSV and PCV are given, the sequence of administration may be important. In older adults, prior administration of PPSV23 blunts the opsonophagocytic antibody response to subsequent administration of PCV13 [29]. Nonetheless, this blunted antibody response may be overcome using a double-dose of PCV13. In a randomized trial among 595 adults aged 55–74 years, a double dose of PCV13 following PPSV23 had superior immune response compared to a single dose of PCV13 following PPSV23 [30]. Table 1 summarizes the different pneumococcal vaccines available, including their serotype coverage, recommended age groups, and dosing schedules.

## 5. Immunogenicity and Clinical Efficacy of Pneumococcal Vaccination

The protective efficacy of PPSV23 has been consistently shown by several trials [33,34,35], and in a meta-analysis of 18 randomized trials including >64,500 individuals [36]. For PCV13, in a large randomized, double-blind, placebo-controlled trial involving 84,496 adults aged ≥65 years, PCV13 was shown to prevent about half of vaccine-type pneumococcal pneumonia occurrences (49 vs. 90 events) [37]. Efficacy persisted for the duration of the trial, with a mean follow-up of four years.

Given the huge expense required to run large vaccine efficacy trials with relatively small event rates, studies of newer vaccines tend to compare immunogenicity to PCV13, and infer clinical efficacy from immunogenicity (i.e., bridging studies) [38]. Examples of these bridging studies with PCV13 as the comparison vaccine have demonstrated non-inferior immunogenicity of PPSV23 and newer vaccines such as PCV15, PCV20, and PCV24. Non-inferiority was demonstrated in both healthy adults (Table 2) and in patients with chronic medical conditions or immunocompromised states (Table 3).

Little reason exists to avoid pneumococcal vaccination in immunocompromised patients. For instance, PCV15 and PPSV23 are both well tolerated and immunogenic for persons living with human immunodeficiency virus (HIV) [55,56]. PCV7 and PCV13 have previously been shown to also reduce the risk of vaccine-type pneumonia among HIV patients [57,58]. Even among severely immunocompromised patients, continued vaccination may still be supported. For example, while patients with lymphoma or multiple myeloma are unlikely to develop good antibody responses following vaccination with PCV or PPSV [59], pneumococcal vaccination should still be done as the potential benefits of even suboptimal responses greatly outweigh the risks. On a related note, cancer patients treated with chemotherapy should be vaccinated even if chemotherapy has been started, as serologic response to PCV13 given at the start of chemotherapy is comparable to PCV13 given 2 weeks prior to chemotherapy [60].

Among pregnant mothers, PPSV23 may be preferred in HIV-infected mothers, as PCV10 was associated with decreased antibody responses to PCV10 and seroprotection rates in infants who subsequently received PCV10 [61]. Although protective, maternal antibodies interfere with infant responses to homologous childhood vaccines by the blunting of infant antibody production (i.e., negative maternal antibody interference). Furthermore, PPSV23 has broader maternal protection compared to PCV10 [56,62], as the former covers a larger number of serotypes.

## 6. Mucosal Immunity and Indirect Effect

Unlike PPSV, PCV stimulates mucosal immunity and prevents nasal colonization of *S. pneumoniae*. Since successfully vaccinated individuals cannot be infected, they would not be able to spread infection to others. Mucosal immunity thus reduces the potential for human-to-human transmission of *S. pneumoniae*, which causes an indirect effect (i.e., herd protection) at the population level. Vaccination of invasive pneumococcal disease in South Africa was shown to have both direct and indirect effects in the GERMS-SA study [63]. In this national study, the incidence of invasive pneumococcal disease was tracked across periods of PCV7 and PCV13 introduction in infants. 35,192 cases of invasive pneumococcal disease were identified, with large declines of vaccine serotype-specific disease incidence of 89% in children younger than 2 years. Although adults 25–44 years did not receive pneumococcal vaccination, a large drop of vaccine serotype-specific disease incidence of 57% still occurred among these young adults, reflecting an indirect effect of infant pneumococcal vaccination.

Indirect effect of childhood pneumococcal vaccination was also found in the United States, where the routine schedule for PCV (either PCV13 or PCV15) includes a three-dose primary series and a booster dose [31,32]. Using Active Bacterial Core surveillance and the National Health Interview Survey, PCV13-serotype invasive pneumococcal disease incidence fell by 74% in adults aged 19–64 years without any indication for PCV13 or PPSV23 while PPSV23-unique serotype invasive pneumococcal disease incidence increased by 32% [64]. This demonstrated the indirect effect of childhood PCV13 vaccination which would have protected adults against PCV13-specific serotypes but not PPSV23-unique serotypes. As such, in areas where childhood PCV administration has decreased PCV-specific serotype circulation, adults who require pneumococcal vaccination may benefit more from PPSV23 than PCV13 as PPSV23 includes protection against a broader range of serotypes [56,62]. On the other hand, the indirect effect of vaccination cannot be totally relied upon for herd protection, especially if childhood vaccination coverage is poor, if adults travel to regions with low childhood PCV vaccination rates, or if disease transmission patterns involve adult-to-adult spread rather than child-to-adult spread [65].

## 7. Coadministration of Pneumococcal Vaccine with Other Vaccines

Coadministration of pneumococcal vaccines with other vaccines have demonstrated little cross-interference, especially with the inactivated influenza vaccine. Coadministration of PPSV23 and trivalent inactivated influenza vaccine (IIV3) or quadrivalent inactivated influenza vaccine (IIV4) yielded no worsening of antibody responses and no increased adverse reactions in two separate trial populations: 162 adults aged ≥ 65 years [66] and 41 adults with coronary artery disease (mean age 66 years) [67].

PCV15 administered concomitantly with IIV4 is generally well tolerated and immunologically non-inferior to non-concomitant administration, supporting coadministration of both vaccines, according to a trial involving 1200 participants (about 50% aged ≥ 65 years) [68]. Similar immunogenicity and safety were also shown in another trial of 846 adults aged ≥ 50 years, who received PPSV23 ≥ 1 year before [69]. Noninferiority for pneumococcal and influenza antibody responses and similar adverse event rates for coadministration of PCV20 and IIV4, compared to IIV4, then PCV20 1 month later, was found in a further trial of 1796 adults ≥ 65 years of age [70].

Coadministration of pneumococcal vaccine with recombinant zoster vaccine (RZV) has also showed no immunologic interference in a trial of 865 participants ≥ 50 years of age, who were randomly assigned into two groups: Group 1 received coadministration of PPSV23 and recombinant zoster vaccine (RZV), then RZV 2 months later, while Group 2 received PPSV23, then RZV 2 and 4 months later [71]. An analogous comparison in another trial of 912 adults with mean age 62.9–63.2 years (~25% who were ≥70 years of age) again showed noninferiority for PCV13 and RZV antibody responses for coadministration [72].

Though coadministration of pneumococcal vaccine with COVID-19 vaccines would be desirable to improve vaccination coverage, published data on immunologic interference are not available [73]. Such data should gradually be available, given a preliminary report from a Phase III randomized trial (ClinicalTrials.gov Identifier NCT04887948, last updated 14 December 2022) supporting the safety and immunogenicity of PCV20 co-administered with a booster dose of mRNA COVID-19 vaccine.

Nevertheless, coadministration of pneumococcal vaccines with tetanus-diphtheria vaccine (Td) may not be desirable. In a trial of 448 adults aged ≥ 50 years, the coadministration of PCV13 with Td lead to similar immunogenicity and safety for PCV13 but decreased immunogenicity for Td when co-administered [74]. On a similar note, coadministration of PPSV23 and live-attenuated varicella-zoster vaccine (as opposed to recombinant zoster vaccine) may also not be suitable, as coadministration reduced humoral response to PPSV23 among 52 older patients with diabetes mellitus (mean age 65.8–66.7 years) [75].

## 8. Waning Immunity and Replacement Strains

Administration of additional pneumococcal vaccines among adults may be required due to waning immunity and the emergence of replacement strains, both of which erode the protection afforded by existing pneumococcal vaccines. For PPSV23, effectiveness wanes over several years [19,76], and necessitates consideration of revaccination every 5–10 years for patients at continued risk of severe disease. In a study of older adults aged 75 years and above, PPSV23 vaccine effectiveness dropped from 74% in the first year to 15% in the fifth year [77]. Similarly, in another study of adults aged 65 years and above, PPSV23 vaccine effectiveness decreased from 41% within two years of vaccination to 23% five years after vaccination [78]. In contrast, PCV13 vaccine effectiveness does not seem to wane at least over 4 years post-vaccination [79], though another follow-up study of PCV7 showed considerable waning of PCV7-serotype specific opsonophagocytic activity over 6 years [80]. Longer-term studies are therefore required to clarify waning immunity of various types of PCV.

Emergence of replacement strains (i.e., emergence of nonvaccine serotypes that can similarly cause invasive pneumococcal disease) creates an epidemiological moving target for pneumococcal vaccination, necessitating development of vaccines that cover new serotypes or that do not depend on specific capsular polysaccharides for protection. For example, a few years after universal vaccination with PCV7 commenced in the United States, *S. pneumoniae* type 19A (not included in PCV7) became the commonest cause of pneumococcal disease in both children and adults. Similarly, for PCV13, non-vaccine serotypes have increased in prevalence since PCV13 was introduced [81,82,83].

The clinical impact of replacement strains was demonstrated in the GERMS-SA study, which investigated the direct and indirect effects of pneumococcal vaccination [63]. Among children not infected with the human immunodeficiency virus, while the incidence of vaccine serotype-related invasive pneumococcal disease dropped by 85%, the incidence of disease caused by nonvaccine serotypes increased by 33% over the same period of observation. On the same note, from a systematic review of 29 observational studies and 2,033,961 cases, childhood PCV13 vaccination led to a significant increase in non-vaccine type invasive pneumococcal disease among adults, especially in those aged over 65 years [84]. Emergence of replacement strains highlights the need for continued surveillance at the population level and suggests that new extended-valency pneumococcal vaccines covering a broader array of serotypes are needed at an individual level, even in persons who were previously vaccinated with the older pneumococcal vaccines.

## 9. Future Directions

Two areas of adult pneumococcal vaccination require further research. Firstly, vaccination strategies with the existing vaccines require clarification. It remains uncertain how vaccine immunogenicity and efficacy will wane over the long-term in either healthy or immunocompromised patients. These will have implications for revaccination intervals. Furthermore, for immunocompromised patients who have blunted immune responses [59], vaccination strategies such as doubling the dose of existing vaccines seem to work but need validation. For example, 65 kidney transplant (TC) candidates and 74 transplant recipients (TR) (mean age 50–54.5 years) were randomized to a double dose (DD) group (DD PCV13, then DD PPSV23 12 weeks later) and a normal dose (ND) group (normal dose (ND) PCV13, then ND PPSV23 12 weeks later). Protective response (geometric mean concentration ≥ 1 mg/L, averaged across 12 vaccine-shared-serotype IgG) 5 weeks post-PPSV23 was higher in DD-TC than ND-TC (71% vs. 37%, *p* = 0.008), but similar in TR [85,86]. Nonetheless increased dosing is not always needed. Among 70 allogeneic hematopoietic stem cell transplant recipients (mean age 47.2 ± 14.4 years), 4 versus 3 doses PCV13 before PPSV23 did not yield a significant difference in the IgG response rate (≥0.20 mg/mL) for 8 measured serotypes at 5 months after the PPSV23 booster (100% vs. 93%) [87].

Secondly, new vaccines are needed to mitigate the emergence of replacement strains. Existing vaccines that use capsular polysaccharides as antigens are limited by the number of serotypes covered. A novel vaccine development approach taps on highly conserved pneumococcal proteins (e.g., pneumolysin, histidine triad protein D, surface proteins A and C) [88,89,90,91], which are common to essentially all serotypes. Such a vaccine can then broadly target all *S. pneumoniae* serotypes (i.e., serotype-independent protection). However, for unclear reasons, a recent clinical trial of a protein-based pneumococcal vaccine co-administered with PCV13 in infants did not demonstrate incremental efficacy in preventing acute otitis media or acute lower respiratory tract infection [92]. Non-serotype-specific vaccine protection can also be provided using an inactivated *S. pneumoniae* whole cell vaccine, which relies on killed cells from a nonencapsulated strain of *S. pneumoniae*, coupled with an aluminium hydroxide adjuvant. A whole cell vaccine could potentially protect against more diverse serotypes than existing capsular polysaccharide-based vaccines [93]. In a randomized trial of 42 healthy adults (mean age 25.2–29.9 years), *S. pneumoniae* whole cell vaccine (3 doses 4 weekly) was well-tolerated and induced pneumococcal antigen-specific antibody and T-cell cytokine responses [94].

## 10. Conclusions

Guidelines currently recommend pneumococcal vaccination among patients at risk of severe disease, identifying age and comorbidity as the main risk factors. Decision-making for individuals additionally require clinicians to consider risk of exposure to infection, immunogenicity, clinical efficacy, mucosal immunity, herd immunity, concomitant administration with other vaccines, waning immunity, and replacement strains (Table 4). In general, pneumococcal vaccination is recommended for patients at risk of severe disease, covering as many serotypes as possible. Coadministration with other vaccines like the inactivated influenza vaccine may be done if safety and immunogenicity are preserved. Looking ahead, vaccination strategies with the existing vaccines require clarification regarding revaccination intervals and improving immunogenicity in immunocompromised patients. Additionally, novel vaccines are needed to mitigate the emergence of replacement strains, which present an epidemiological moving target for existing capsular polysaccharide-based vaccines.

## Figures and Tables

**Table 1 vaccines-11-00908-t001:** Summary of available pneumococcal vaccines.

Vaccine [Ref]	Brand Name	Serotype Coverage	Recommended Age Groups	Dosing Schedules
PCV13 [31]	Prevnar 13 (Pfizer, Inc., New York, NY, US)	1, 3, 4, 5, 6A, 6B, 7F, 9V, 14, 18C, 19A, 19F, 23F	All infants and children 2–59 months of age Children * and adults ** ≥ 60 months of age with underlying medical conditions Adults ≥ 65 years of age	Infants ≥ 6 weeks and children ≤ 15 months: IM 0.5 mL per dose given at 2, 4, 6, and 12 through 15 months of age (for a total of 4 doses) Adults IM 0.5 mL as a single dose
PCV15 [10,32]	Vaxneuvance (Merck and Co, Inc., Rahway, NJ, US)	22F and 33F, in addition to PCV13 serotypes	All infants and children 2–59 months of age Children * and adults ** ≥60 months of age with underlying medical conditions Adults ≥65 years of age	Infants ≥6 weeks and children ≤15 months: IM 0.5 mL per dose given at 2, 4, 6, and 12 through 15 months of age (for a total of 4 doses) Adults IM 0.5 mL as a single dose
PCV20 [10]	Prevnar 20 (Pfizer, Inc., New York, NY, US)	8, 10A, 11A, 12F, 15B, 22F, and 33F, in addition to PCV13 serotypes	Adults 19–64 years of age with underlying medical conditions ** Adults ≥ 65 years of age	Adults IM 0.5 mL as a single dose If PCV20 is given, then PPSV23 is not required
PPSV23 [31]	Pneumovax 23 (Merck and Co, Inc., Rahway, NJ, US)	2, 8, 9N, 10A, 11A, 12F, 15B, 17F, 20, 22F, 33F, in addition to PCV13 serotypes, less 6A	Children * and adults ** ≥ 60 months of age with underlying medical conditions Adults ≥65 years of age	IM or SC 0.5 mL as a single dose Administer PPSV23 ≥ 1 year after PCV13 or PCV15 Administer a final dose of PPSV23 ≥ 5 years after the previous dose of PPSV23 once the patient turns 65 years of age

* Immunocompetent children with chronic heart disease (particularly cyanotic congenital heart disease and heart failure), chronic lung disease (including asthma if treated with high dose corticosteroids), diabetes, cerebrospinal fluid leaks, or cochlear implants; children with functional or anatomic asplenia, including sickle cell disease or other hemoglobinopathies, congenital or acquired asplenia, or splenic dysfunction; children with immunocompromising conditions including congenital immunodeficiency (includes B- or T-cell deficiency, complement deficiencies and phagocytic disorders), HIV infection, chronic renal failure, nephrotic syndrome, leukemia, lymphoma, Hodgkin disease, generalized malignancies, solid organ transplant, or other diseases requiring immunosuppressive drugs (including long-term systemic corticosteroids and radiation therapy). ** Alcoholism, chronic heart disease (including heart failure, cardiomyopathies), chronic liver disease, chronic lung disease (including chronic obstructive pulmonary disease, emphysema, asthma), cigarette smoking, diabetes mellitus, cochlear implant, cerebrospinal fluid leaks, asplenia (congenital or acquired), sickle cell disease or other hemoglobinopathies, immunocompromising conditions (e.g., chronic renal failure, congenital or acquired immunodeficiency [including B- or T-cell deficiency, complement deficiencies and phagocytic disorders;], malignancy, HIV infection, Hodgkin disease, iatrogenic immunosuppression [including long-term systemic corticosteroid treatment, radiation therapy], leukemia, lymphoma, multiple myeloma, nephrotic syndrome, solid organ transplant). IM: Intramuscular; SC: Subcutaneous.

**Table 2 vaccines-11-00908-t002:** Bridging studies of pneumococcal vaccination in healthy adults.

Author (Year) [Ref]	Population	Intervention	Comparison	Outcome
Ermlich (2018) [39]	691 adults aged ≥ 50 years	PCV15	PCV13 or PPSV23	Compared to PCV13 and PPSV23, PCV15 had better immunogenicity for shared serotypes and similar safety at 1-month post-vaccination
Kishino (2022) [40]	245 Japanese adults ≥ 65 years of age	PCV15 (includes PCV13 serotypes, and serotypes 22F and 33F)	PCV13	Higher proportion of participants with a ≥4-fold increase in serotype-specific opsonophagocytic activity responses for PCV15 than for PCV13 for serotypes 3, 22F, and 33F
Peterson (2019) [41]	250 adults ≥ 65 years of age with prior PPSV23 ≥ 1 year before	PCV15	PCV13	Good safety and immunogenicity in both trial groups
Platt (2022) [42]	1202 adults (about 30% aged 50–64 years)	PCV15	PCV13	PCV15 non-inferior to PCV13 for 13 shared serotypes, and superior immunogenicity for the 2 unique serotypes
Simon (2022) [43]	1299 healthy adults ≥ 65 years of age	PCV15, then PPSV23 6 months later	PCV13, then PPSV23 6 months later	Similar immunogenicity and safety
Song (2021) [44]	652 adults (about 50% aged ≥ 65 years)	PCV15, then PPSV23 12 months later	PCV13, then PPSV23 12 months later	Similar immunogenicity and safety
Stacey (2019) [45]	690 adults ≥ 50 years of age	PCV15	PCV13	Similar immunogenicity and safety
Cannon (2021) [46]	875 adults ≥ 65 years of age, with prior pneumococcal vaccination	PCV20	PCV13 (if prior PPSV23); PPSV23 (if prior PCV13)	PCV20 was immunogenic 1 month after PCV20, regardless of prior PCV13 or PPSV23
Essink (2022) [47]	893 generally healthy pneumococcal vaccine-naïve adults 18–59 years of age	PCV20	PCV13	PCV20 was safe and well tolerated. Immunogenicity of PCV20 comparable to that of PCV13
Essink (2022) [47]	3009 generally healthy pneumococcal vaccine-naïve adults ≥ 60 years of age	PCV20, then PPSV23 1 month later	PCV13, then PPSV23 1 month later	PCV20 was safe and well tolerated. Immunogenicity of PCV20 comparable to that of PCV13 or PPSV23
Fitz-Patrick (2021) [48]	104 healthy Japanese adults 18–49 years of age residing in the United States for ≤5 years	PCV20, or complementary 7-valent PCV (cPCV7) (containing non-PCV13 serotypes)	PCV13	Good opsonophagocytic activity at 1-month post-vaccination
Klein (2021) [49]	1710 adults 18–49 years of age	PCV20	PCV13	Similar immunogenicity and safety
Hurley (2021) [26]	444 adults 60–64 years of age	PCV20, then placebo 1 month later	PCV13, then PPSV23 1 month later	Similar immunogenicity and safety
Chichili (2022) [28]	126 pneumococcal vaccine-naïve adults 18–64 years of age. 390 pneumococcal vaccine-naïve adults 65–85 years of age	PCV24	PCV13	Higher proportion of participants with opsonophagocytic activity responses at day 30 for PCV24 than for PCV13 for both PCV13 and non-PCV13 serotypes

**Table 3 vaccines-11-00908-t003:** Bridging studies of pneumococcal vaccination in adult patients with chronic medical conditions or immunocompromised states.

Author (Year) [Ref]	Population	Intervention	Comparison	Outcome
Eriksson (2020) [50]	74 adult kidney transplant recipients, mean age 52.4–55.9 years	PPSV23 before transplant	PCV13 before transplant, with repeat PCV13 6 months after	Similar immunogenicity and safety. Waning sero-response post-transplant restored with repeat PCV13
Eriksson (2021) [51]	47 adult liver transplant recipients, mean age 54.9–56.1 years	PPSV23 before transplant	PCV13 before transplant, with repeat PCV13 6 months after	Similar immunogenicity and safety. Waning sero-response post-transplant restored with repeat PCV13
Kantso (2019) [52]	82 Crohn’s disease patients with or without immunosuppressive therapy, mean age 44 ± 14 years	PPSV23	PCV13	Similar persistence of induced antibodies in both trial groups, with similarly reduced persistence when patients were on combination thiopurines and TNF-α antagonists
Hammit (2022) [53] (2023) [54]	1131 at-risk adults * and 381 adults without risk factors 18–49 years of age	PCV15, then PPSV23 6 months later	PCV13, then PPSV23 6 months later	Similar immunogenicity, safety, and tolerability across risk factor groups
Mohapi (2022) [55]	302 adults living with HIV and receiving antiretroviral therapy, mean age 41.3–42.4 years (range 21–74 years)	PCV15, then PPSV23 8 weeks later	PCV13, then PPSV23 8 weeks later	Similar opsonophagocytic activity geometric mean titers for shared serotypes at day 30 and week 12

* Risk factors include one or more of the following: (1) Diabetes mellitus type 1 or 2, receiving anti-diabetic medication; glycated hemoglobin < 10% at screening: (2) Chronic liver disease with compensated cirrhosis (Child-Pugh Class A) due to nonalcoholic fatty liver disease, chronic hepatitis B or C, or alcoholic liver disease, with at least one liver staging assessment; (3) Chronic obstructive pulmonary disease with ratio of forced expiratory flow in the 1st second divided by the forced vital capacity (FEV1/FVC) <0.7, and FEV1 ≥30% predicted in the prior 5 years; (4) Mild or moderate persistent asthma with reversible airflow obstruction on spirometry and receipt of guideline-directed therapy for mild-to-moderate asthma; (5) Chronic heart disease due to heart failure with reduced or preserved ejection fraction or non-cyanotic congenital heart disease, diagnosed in the prior 5 years and classified as New York Heart Association heart failure Class 1–3 with receipt of guideline-directed oral heart failure treatment; (6) Current smoker (≥100 cigarettes during lifetime) and not currently receiving smoking cessation therapy; (7) Alcoholism defined as an Alcohol Use Disorders Identification Test (AUDIT-C) score of ≥5.

**Table 4 vaccines-11-00908-t004:** Individualized decision-making for pneumococcal vaccination in adults.

Clinical Question	Considerations	Details
Should the patient receive pneumococcal vaccination?	Risk of severe pneumococcal disease	Yes, for patients at risk of severe invasive or non-invasive pneumococcal disease, which is in turn dependent on patient factors (age, comorbid conditions), and on exposure risk (epidemiology)
Which pneumococcal vaccine or vaccines should be used?	Immunogenicity Clinical efficacy Mucosal immunity Indirect effect Herd protection	Chosen vaccine should be immunogenic and safe, with as broad protection as possible to cover pathogenic serotypes. Chosen vaccine should have demonstrated reduction of adverse health outcomes such as hospitalization or mortality. Immunocompromised patients benefit despite having lower vaccine immune responses. Consider an additional pneumococcal vaccine if broadened serotype coverage can help prevent severe invasive or non-invasive pneumococcal disease, especially when herd protection is insufficient
Can pneumococcal vaccine be co-administered with other vaccines?	Immunologic interference Safety and tolerability	Yes, if immunogenicity of both vaccines is not affected and if adverse events are not significantly increased, e.g., with inactivated influenza vaccines
Should pneumococcal vaccination be repeated?	Waning immunity Replacement strains	Yes, if waning immunity is expected. Immunocompromised patients may benefit from repeated vaccination due to lower vaccine immunogenicity. Emergence of replacement strains may necessitate broadened serotype cover by administering a different vaccine, one that may already be existing or a new vaccine that becomes available to cover additional pathogenic serotypes

## Data Availability

All data used can be found in the text and tables.

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
