# Peer review of "Pneumococcal Vaccination in Adults: A Narrative Review of Considerations for Individualized Decision-Making"

_vaccines, 2023, doi:10.3390/vaccines11050908_

Round 1

Reviewer 1 Report

In this review the author has described the considerations for individual decision-making for pneumococcal adult vaccination. These considerations include risk of severe disease, immunogenicity, clinical efficacy, mucosal immunity, herd immunity, concomitant administration with other vaccines, waning immunity, and replacement strains.

This review has been well structured and written. 

In the Future Directions chapter, the author suggested that new vaccines are needed to mitigate the emergence of replacement strains.

'Existing vaccines that use capsular polysaccharides as antigens are limited by the number of serotypes covered. A novel vaccine development approach taps on highly conserved pneumococcal proteins (e.g., pneumolysin, histidine triad protein D, surface proteins A and C) [87-90], which are common to essentially all serotypes. Such a vaccine can then 301 broadly target all S. pneumoniae serotypes.'

Please author expand the text to highlight that protein based vaccines tested in clinic against pneumococcal infections unfortunately failed.  

No 

Author Response

In this review the author has described the considerations for individual decision-making for pneumococcal adult vaccination. These considerations include risk of severe disease, immunogenicity, clinical efficacy, mucosal immunity, herd immunity, concomitant administration with other vaccines, waning immunity, and replacement strains.

This review has been well structured and written.

Reply: Thank you for your kind comments.

In the Future Directions chapter, the author suggested that new vaccines are needed to mitigate the emergence of replacement strains.

'Existing vaccines that use capsular polysaccharides as antigens are limited by the number of serotypes covered. A novel vaccine development approach taps on highly conserved pneumococcal proteins (e.g., pneumolysin, histidine triad protein D, surface proteins A and C) [87-90], which are common to essentially all serotypes. Such a vaccine can then 301 broadly target all S. pneumoniae serotypes.'

Please author expand the text to highlight that protein based vaccines tested in clinic against pneumococcal infections unfortunately failed.

Reply: Updated “Such a vaccine can then broadly target all S. pneumoniae serotypes” to “Such a vaccine can then broadly target all S. pneumoniae serotypes (i.e., serotype-independent protection). However, for unclear reasons, a recent clinical trial of a protein-based pneumococcal vaccine co-administered with PCV13 in infants did not demonstrate incremental efficacy in preventing acute otitis media or acute lower respiratory tract infection [92].”

Reviewer 2 Report

The article under review is well structured and clearly written. It has a well-developed introductory part. The summary is clear and synthetic. I have two remarks: The selection of the analyzed articles is not described (what were the selection criteria, what databases were used and what keywords were used). Supplementing this information will strengthen the methodological aspect of the work. Are there any articles indicating the need to take into account the fact of COVID-19 when making individual decisions?

Author Response

The article under review is well structured and clearly written. It has a well-developed introductory part. The summary is clear and synthetic. I have two remarks: The selection of the analyzed articles is not described (what were the selection criteria, what databases were used and what keywords were used). Supplementing this information will strengthen the methodological aspect of the work.

Reply: Added to Methods “A search of Pubmed® (pubmed.ncbi.nlm.nih.gov) of contemporary trials over the past 5 years from 2018-2023 was performed using the term “pneumococcal vaccination randomized trials”. This was done to update the author's personal library of articles. Articles relevant to the considerations covered in this narrative review were included.”

Are there any articles indicating the need to take into account the fact of COVID-19 when making individual decisions?

Reply: Added to 7. Coadministration of pneumococcal vaccine with other vaccines “Though coadministration of pneumococcal vaccine with COVID-19 vaccines would be desirable to improve vaccination coverage, published data on immunologic interference are not available [73]. Such data should gradually be available, given a preliminary report from a Phase III randomized trial (ClinicalTrials.gov Identifier NCT04887948, last updated 14 Dec 2022) supporting the safety and immunogenicity of PCV20 coadministered with a booster dose of mRNA COVID-19 vaccine.”

Reviewer 3 Report

Reviewer Report with Minor Revisions

Manuscript ID: vaccines-2349366 Title: Pneumococcal vaccination in adults: Considerations for individualized decision-making Section: Vaccines against Infectious Diseases Special Issue: Communicable Diseases: New and Old Therapy and Preventive Strategies

General Comments: The manuscript provides a comprehensive overview of the considerations for individualized decision-making for pneumococcal vaccination in adults. The writing is clear, and the structure is well-organized. Overall, the manuscript is suitable for publication in Vaccines, pending minor revisions.

Specific Comments:

  1. The introduction section can be further strengthened by providing more background information on pneumococcal disease, such as its incidence, mortality, and morbidity rates. This will help readers to better understand the importance of pneumococcal vaccination in adults.
  2. It would be helpful to provide a table summarizing the different pneumococcal vaccines available, including their serotype coverage, recommended age groups, and dosing schedules.
  3. In the discussion section, the authors mention the potential use of biomarkers to identify individuals who are at higher risk for pneumococcal disease. It would be helpful to provide some examples of such biomarkers and their potential clinical applications.
  4. The manuscript would benefit from a more thorough proofreading for grammatical errors and typos.

Minor Revisions:

  1. In the abstract, the sentence "These considerations include risk of severe disease, immunogenicity, clinical efficacy, mucosal immunity, herd immunity, concomitant administration with other vaccines, waning immunity, and replacement strains" can be modified to "This review identifies and discusses considerations for individualized decision-making, including the risk of severe disease, immunogenicity, clinical efficacy, mucosal immunity, herd immunity, concomitant administration with other vaccines, waning immunity, and replacement strains."
  2. In the conclusion section, the authors can consider providing a summary of the main findings of the review and their implications for clinical practice.
  3. The authors should proofread the manuscript carefully to ensure that there are no grammatical errors or typos.

Overall, the manuscript is well-written and provides valuable insights into the considerations for individualized decision-making for pneumococcal vaccination in adults. I recommend that the manuscript be accepted for publication in Vaccines, pending minor revisions.

See the report.

Author Response

General Comments: The manuscript provides a comprehensive overview of the considerations for individualized decision-making for pneumococcal vaccination in adults. The writing is clear, and the structure is well-organized. Overall, the manuscript is suitable for publication in Vaccines, pending minor revisions.

Reply: Thank you for your kind comments.

Specific Comments:

The introduction section can be further strengthened by providing more background information on pneumococcal disease, such as its incidence, mortality, and morbidity rates. This will help readers to better understand the importance of pneumococcal vaccination in adults.

Reply: Updated in the Introduction from "Pneumococcal disease remains one of the major causes of severe disease in both children and adults, especially when invasive pneumococcal disease (i.e., isolation of S. pneumoniae from a normally sterile site, and associated with bacteremic pneumonia, meningitis, endocarditis, or arthritis) or severe nonbacteremic pneumonia occur [4]. Such severe pneumococcal infections cause substantial morbidity and mortality [5]. Additionally, S. pneumoniae is the leading bacterial cause of pneumonia globally [6], and is responsible for about a third of bacterial community-acquired pneumonia in the United States [7]" to “Pneumococcal disease remains one of the major causes of severe disease in both children and adults, especially when invasive pneumococcal disease (i.e., isolation of S. pneumoniae from a normally sterile site, and associated with bacteremic pneumonia, meningitis, endocarditis, or arthritis) or severe nonbacteremic pneumonia occur. Annually, in the United States, invasive pneumococcal disease has an incidence of 10.6/100,000, while pneumococcal pneumonia accounts for 900,000 cases and 400,000 hospitalizations [4]. Such severe pneumococcal infections cause substantial morbidity and mortality, with mortality from pneumococcal pneumonia ranging from 5 to 7% [4,5]. Additionally, S. pneumoniae is the leading bacterial cause of pneumonia globally [6], and is responsible for about a third of bacterial community-acquired pneumonia in the United States [7].”

It would be helpful to provide a table summarizing the different pneumococcal vaccines available, including their serotype coverage, recommended age groups, and dosing schedules.

Reply: Included a new Table 1. Summary of available pneumococcal vaccines.

In the discussion section, the authors mention the potential use of biomarkers to identify individuals who are at higher risk for pneumococcal disease. It would be helpful to provide some examples of such biomarkers and their potential clinical applications.

Reply: Added to 3. Risk of severe pneumococcal disease in adults "Immunocompromised states increase the risk of pneumococcal disease, and these states can be identified using biomarkers such as neutropenia, low CD4 cell count, and hypogammaglobulinemia. Functional or anatomic asplenia due to sickle cell disease or splenectomy predisposes to recurrent pneumococcal disease."

The manuscript would benefit from a more thorough proofreading for grammatical errors and typos.

Reply: Further proofreading has been done.

Minor Revisions:

In the abstract, the sentence "These considerations include risk of severe disease, immunogenicity, clinical efficacy, mucosal immunity, herd immunity, concomitant administration with other vaccines, waning immunity, and replacement strains" can be modified to "This review identifies and discusses considerations for individualized decision-making, including the risk of severe disease, immunogenicity, clinical efficacy, mucosal immunity, herd immunity, concomitant administration with other vaccines, waning immunity, and replacement strains."

Reply: The abstract has been modified as suggested.

In the conclusion section, the authors can consider providing a summary of the main findings of the review and their implications for clinical practice.

Reply: Added to conclusion “In general, pneumococcal vaccination is recommended for patients at risk of severe disease, covering as many serotypes as possible. Coadministration with other vaccines like the inactivated influenza vaccine may be done if safety and immunogenicity are preserved.”

The authors should proofread the manuscript carefully to ensure that there are no grammatical errors or typos.

Reply: Further proofreading has been done.

Overall, the manuscript is well-written and provides valuable insights into the considerations for individualized decision-making for pneumococcal vaccination in adults. I recommend that the manuscript be accepted for publication in Vaccines, pending minor revisions.